# Examination of acute spin exercise on GABA levels in aging and stroke: The EASE study protocol

Keith M. McGregor [1,2]*, Thomas Novak[3,4,5], Joe R. Nocera[3,4,5], Kevin Mammino[3], Steven L. Wolf[3,5,6,7], Lisa C. Krishnamurthy[8,9,10,11]

1 Birmingham VA Geriatric Research Education and Clinical Center, Birmingham VA Health Care System, Birmingham, Alabama, United States of America, 2 Department of Clinical and Diagnostic Sciences, University of Alabama at Birmingham School of Health Professions, Birmingham, Alabama, United States of America, 3 Rehabilitation R&D Center for Visual and Neurocognitive Rehabilitation, Atlanta VA Health Care System, Atlanta, Georgia, United States of America, 4 Department of Neurology, Emory University School of Medicine, Atlanta, Georgia, United States of America, 5 Division of Physical Therapy, Department of Rehabilitation Medicine, Emory University School of Medicine, Atlanta, Georgia, United States of America, 6 Department of Medicine, Emory University School of Medicine, Atlanta, Georgia, United States of America, 7 Department of Cell Biology, Emory University School of Medicine, Atlanta, Georgia, United States of America, 8 Atlanta VA Health Care System, Decatur, Georgia, United States of America, 9 Tri-Institutional Center for Translational Research in Neuroimaging and Data Science (TReNDS), Georgia State, Georgia Tech and Emory, Atlanta, Georgia, United States of America, 10 Department of Physics & Astronomy, Georgia State University, Atlanta, Georgia, United States of America, 11 Department of Radiology and Imaging Sciences, Emory University, Atlanta, Georgia, United States of America

* kmmcgreg@uab.edu

**Data Availability Statement:** No datasets were generated or analysed during the current study. All

# Abstract

## Background

Changes in regional levels of the inhibitory neurotransmitter gamma-aminobutyric acid (GABA) may indicate the potential for favorable responses to the treatment of stroke affecting the upper extremity. By selectively altering GABA levels during training, we may induce long-term potentiation and adjust excitatory/inhibitory balance (E/I balance). However, the impact of this alteration may be limited by neural damage or aging. Aerobic exercise has been shown to increase GABA levels in the sensorimotor cortex and improve motor learning by widening the dynamic range of E/I balance. The cross-sectional project, Effects of Acute Exercise on Functional Magnetic Resonance Spectroscopy Measures of GABA in Aging and Chronic Stroke (EASE), is designed to assess the functional relevance of changes in GABA concentration within the sensorimotor cortex before and after an acute aerobic exercise session.

## Methods/design

EASE will enroll 30 participants comprised of healthy younger adults (18–35 years; n = 10), older adults (60+ years; n = 10), and persons with chronic stroke (n = 10) affecting distal upper extremity function. We will use resting magnetic resonance spectroscopy to measure all participants' GABA levels at rest before and after aerobic exercise. In addition, we will employ functional magnetic resonance spectroscopy using motor skill acquisition and recall

relevant data from this study will be made available upon study completion.

**Funding:** This study is supported by the Department of Veterans Affairs Small Projects in Rehabilitation Award (SPiRE: I21RX003581) to KMM, LCK. Funding sources are not involved in the study design, data collection, analysis, or writing of manuscripts. All original data are property of the United States Government through the Department of Veterans Affairs.

**Competing interests:** The authors have declared that no competing interests exist.

**Abbreviations:** FEW, family wise error; fMRI, functional magnetic resonance imaging; fMRS, functional magnetic resonance spectroscopy; FWHM, full width at half maximum; GABA, gamma-aminobutyric acid; GLM, general linear model; GLX, Glutamate/Glutamine; HIIT, High Intensity Interval Training; IPAQ, International Physical Activity Questionnaire; MEGA-PRESS, Mescher-Garwood Point Resolved Spectroscopy; MoCA, Montreal Cognitive Assessment; MNI, Montreal Neurological Institute; PAR-Q+, new Physical Activity Readiness Questionnaire; RPE, Borg Rating of Perceived Exertion; TR, repetition time in magnetic resonance imaging; TE, echo time in magnetic resonance imaging; VO2max, maximal oxygen uptake; %HRmax, percentage of maximal heart rate.

tasks in healthy adults. We hypothesize that acute aerobic exercise will increase resting sensorimotor GABA concentration and that higher GABA resting levels will predict better motor learning performance on measures taken both inside and outside the magnet. We also hypothesize that a higher dynamic range of GABA during task-based spectroscopy in healthy adults will predict better motor skill acquisition and recall.

## Discussion

The EASE project will evaluate the effect of acute exercise on GABA levels as a biomarker of upper extremity motor skill learning with two populations (aging adults and those with chronic stroke). We predict that acute exercise, higher sensorimotor GABA levels, and broader dynamic range will be related to better motor skill acquisition.

## Introduction

### Background

Every year in the United States, approximately 795,000 people experience a stroke; out of that number, 610,000 cases are first-time strokes [1]. Although more people are surviving strokes than before due to advancements in acute care settings, recovery of function remains challenging, and up to 30% of stroke survivors experience severe disability, while another 40% experience functional impairments that require assistance with activities of daily living [2, 3]. Stroke rehabilitation is moving towards personalized rehabilitation therapies that utilize diagnostic biomarkers to customize treatment decisions. Traditional motor rehabilitation therapies that require repetitive movements are time-intensive and have yielded variations in functional outcomes [4–6]. However, more recent approaches should address individual variability to promote the reclamation of function and improve quality of life [7–9]. Incorporating adjuvant therapies, such as aerobic exercise, produces significant benefits on motor outcomes after stroke [10]; yet few studies have identified the mechanisms by which exercise may benefit motor skill acquisition [11, 12]. This study aims to evaluate changes in neurotransmitter concentration, specifically gamma-aminobutyric acid (GABA), as a potential mechanism contributing to how exercise improves motor skill acquisition in younger and older neurologically intact adults and persons with chronic stroke.

### GABA: A target to promote neural plasticity in stroke recovery with exercise

GABA is the brain's primary inhibitory neurotransmitter and has been implicated in both long-term potentiation and long-term depression [13, 14], in effect playing a significant role in inhibitory plasticity [15]. Studies in rodent models have shown that GABA increases in the regions surrounding the locus of the damage after stroke [16–18], likely preventing hypoxic stress and excitotoxicity [16–18]. A recent study showed that GABA is reduced throughout the brain after acute stroke and correlated with upper-extremity Fugl-Meyer scores [18] [NO_PRINTED_FORM]. In the only treatment study to investigate GABA as a neural correlate, Blicher et al. also showed that GABA is reduced after stroke and that constraint-induced movement therapy changes in GABA concentration correlated with improvements in the motor system [19]. Indeed, stroke effects on GABA likely impact rehabilitation efficacy because GABA is involved in learning [20–24], memory [15], and brain plasticity [15], all critical drivers of rehabilitation success after stroke [25, 26]. Changes in GABA concentration due

to acute exercise may potentiate motor learning, but we are not aware of any studies investigating this mechanism.

## Aging, exercise, and neurometabolites

Age strongly predicts response to therapy after a stroke, with older stroke survivors tending to improve less after therapy than younger individuals [27, 28]. The decline in response to motor rehabilitation in older adults is attributed to aging-related decreases in the rate of motor skill acquisition [29–32]. However, recent studies have shown that acute aerobic exercise can improve the rate of motor skill learning in both younger and older adults [33–36]. While the neural mechanisms associated with motor learning benefits of acute exercise are not well known, recent advancements in magnetic resonance spectroscopy (MRS) have shown that aging-related decreases in the concentration of GABA in the sensorimotor cortex may affect cortical excitability [37–39], which is linked to motor learning [40]. Furthermore, studies have shown that individuals with higher resting GABA levels may demonstrate better motor learning during a finger sequence task, and the degree of GABA response during a motor task predicted performance better in older adults than younger adults [22]. Therefore, individualized measures of GABA concentration may predict better learning in stroke survivors [19]. However, the effects of acute exercise on motor learning in relation to resting GABA concentration have not been well explored. Evaluating GABA levels before and after acute exercise in older adults provides an opportunity to evaluate potential metabolic changes due to aging that may be related to functional performance in persons with stroke.

## Current study

In the context of stroke and aging-related GABA system changes, we posit a model that successful learning depends on both 1) the resting concentration of GABA [2, 22] the functional flexibility (or change in focal concentration) of GABA during the task [41–46]. Functional alteration of GABA levels may strongly influence how changes in cortical excitability afford long-term potentiation (LTP). Identifying how GABA concentration changes during motor learning could enable targeting this neurotransmitter (through engagement in aerobic exercise) to facilitate skill acquisition. Measuring GABA at rest will help identify stroke-related perturbations of the resting cortical GABA system, while measuring GABA changes during task-specific behaviors will help identify the GABA system's flexibility to acquire a skill. With the combined information of resting cortical GABA tone and local GABA system flexibility, further identification of which brain regions should be targeted during rehabilitation to facilitate different learning modalities, including motor, language, and visuospatial skill acquisition, may be possible.

In the present study, we will use MRS to measure GABA levels in neurotypical younger and older adults and persons with chronic stroke before and after an acute aerobic exercise session. Neurotypical adults will also engage in a motor skill acquisition and recall task to measure the dynamic range of GABA. We predict that engagement in acute exercise will increase GABA levels to provide a higher dynamic range for motor learning.

## Aims and hypotheses

**Specific Aim 1: To compare pre-post acute exercise changes in resting levels of sensorimotor GABA between younger and older adults using MRS.** Hypothesis: Younger adults will show a larger increase in GABA in sensorimotor areas potentially indicating a limited capacity for within-session GABA change due to the aging process. Greater change in GABA concentration will be correlated with improved motor learning.

**Specific Aim 2: To compare pre-post acute exercise changes in levels of sensorimotor GABA during motor learning between older and younger adults using fMRS.** Hypothesis: Immediately after acute bouts of aerobic exercise older adults will have higher levels of GABA in SM regions, which will be positively correlated with motor learning. Younger adults will have a higher baseline GABA modulatory capacity as compared to older adults.

**Specific Aim 3: To compare changes in resting sensorimotor GABA levels before and after acute exercise patients with chronic stroke.** Hypothesis: After acute exercise, GABA will increase in sensorimotor regions in comparison to the baseline acquisition.

## Methods

### Study design and setting

In this preliminary study, to verify reports of increases in resting GABA in response to a single bout of exercise [37, 47], we will perform assessments on a cohort of 10 healthy younger adults. To extend the understanding of exercise-induced neurochemical changes into an aging model, we will assess a cohort of 10 healthy older adults. Finally, to test the feasibility of detecting acute exercise-induced neurochemical changes in stroke, we will assess ten patients with chronic upper-extremity disability after stroke. Data collection will include two sessions for the healthy younger/older cohorts and three sessions for the stroke group.

The current study design tests change in GABA levels at rest and during motor skill acquisition using magnetic resonance spectroscopy before and after a single bout of high-intensity interval training. The study will be conducted at the Atlanta VA Medical Center (Decatur, GA, USA) and Emory University's Center for Systems Imaging Core (Atlanta, GA, USA). All participants will provide written informed consent in accordance with the Declaration of Helsinki and the Belmont Report. The study protocol was approved by the Institutional Review Board of Emory University (Identifier: STUDY00001334) and the Research Committee at the Atlanta VA Medical Center (Identifier: 1592168–1). Participants will engage in two sessions in the MRI scanner immediately before and after a high-intensity interval training aerobic exercise session lasting 20 minutes and titrated to an individual's aerobic capacity. Moderate to high-intensity exercise is safe for stroke survivors and is a recommendation by the American Heart Association [48].

### Participants

Participants will be recruited from the local population using advertisements in local newspapers, websites, and other local media. As needed, participants with chronic stroke will also be contacted through existing research registries. Before enrollment, persons interested in the study will be screened according to the inclusion and exclusion criteria detailed in Table 1.

Participants will be recruited from the local population using advertisements in local newspapers, websites, and other local media. Adults will be sedentary by self-report engaging in a maximum of 120 minutes of exercise per week. Participants with chronic stroke will also be contacted through existing research registries, as needed. Prior to enrollment in the study, persons with interest in the study will be screened according to the inclusion and exclusion criteria detailed in **Table 1**.

### Study sessions

After telephone screening eligibility determination, all subjects participate in three separate sessions. See Table 2 for an overview of the study flow. After providing informed consent, participants engage in Session 1, which lasts 2 hours and involves motor, cognitive, and quality of

**Table 1. Inclusion and exclusion criteria of younger, older, and chronic stroke participants.**

| Cohort | Inclusion Criteria | Justification |
|---|---|---|
| **Healthy Younger / Older** | Age: 18–35 / 55–80 | Resting GABA levels are known to change across the lifespan with a high likelihood of showing group differences. |
| | English Speaking | Participants must be able to comprehend verbal instructions and communicate with researchers throughout testing. |
| | Right-hand dominance (Edinburgh Handedness Inventory) (Oldfield, 1971) | All motor dexterity/learning assessments are performed with the right hand. The goal is to mitigate potential behavioral/neurological effects attributed to non-dominant manual control. |
| | Montreal Cognitive Assessment (MoCA) score: > 24 | MoCA scores < 24 is indicative of potential cognitive impairment. |
| | Sedentary: < 120 min voluntary aerobic exercise per week | This study examines aging and exercise effects on GABA, and motor performance outcomes in sedentary individuals. |
| **Stroke** | Age: 18–80 | Although stroke is more likely to affect older individuals, more younger individuals are experiencing strokes. |
| | > 6 Months post-stroke, First-time stroke event | This study examines the feasibility of detecting exercise-induced changes in the neurotransmitter system, even in the chronic stages of stroke. The first-time stroke event is meant to reduce heterogeneity in the stroke cohort. |
| | < 120 min voluntary aerobic exercise per week | This study examines aging and exercise effects on GABA and motor performance outcomes in sedentary individuals. |
| | Right-hand dominance (Edinburgh Handedness Inventory) (Oldfield, 1971) with left-hemisphere stroke | All motor dexterity/learning assessments are performed with the right hand. The goal is to assess the effects of stroke on the dominant hemisphere. |
| | Must be able to follow 2-step commands | Individuals need to be able to comprehend and follow instructions during behavioral assessments and in the MRI scanner environment for safety reasons. |
| | Must be able to perform voluntary wrist flexion: $\geq 10°$ 3x/1min | The motor learning task requires volitional control at the wrist joint, and wrist flexion is a notable movement constrained by stroke-related hemiplegia. |

| Cohort | Exclusion Criteria | Justification |
|---|---|---|
| **All participants (Healthy and stroke)** | • Failure to provide informed consent<br>• Severe speech/hearing disorder, language barrier | Participants may volunteer only after they understand and acknowledge the potential risks/benefits of the study. |
| | • Metal/foreign objects in the body<br>• Severe Claustrophobia<br>• Pregnancy | Contraindicated for MR environment |
| | • Actively taking GABAergic agonist/antagonist medications<br>• Self-reported substance abuse | The goal is to assess exercise-induced changes in neurochemistry, which may be altered in the presence of GABAergic medications or substance abuse |
| | • Significant cognitive impairment or progressive/degenerative neurological disease<br>• Severe psychiatric disorder (bi-polar disorder, schizophrenia) | The goal is to assess brain changes in response to stroke while controlling for other neurological disorders. |
| | • Severe visual impairment | Cognitive/motor assessments require normal or corrected vision. |
| | • Pulmonary disease<br>• Cardiac disease: NYHA Class III or IV congestive heart failure, aortic stenosis, use of cardiac defibrillator, uncontrolled angina | Contraindication for HIIT exercise |
| | • Renal failure<br>• Uncontrolled diabetes<br>• Terminal illness (< 12-month life-expectancy) | Co-morbid diseases that may alter the physiological response to aerobic exercise training |
| **Participants with Stroke** | • Significant inability to extend wrist/fingers limits assessment of motor skill acquisition.<br>• Significant gait impairment | Profound hemiparesis that may interfere with exercise and motor testing |

life assessments. Study Session 2 is performed at the study imaging facility with an exercise bicycle stored in an adjoining equipment room to the MRI scanner. This session lasts 3 hours and begins with acclimatization to the MR environment and task familiarization before pre-exercise MRI/MRS scanning lasting 60 minutes. Participants then perform a single session of aerobic spin exercise for 20 minutes (details below). Immediately after exercise, participants are brought back into the scanner for a second MRI/MRS session. Participants with stroke

**Table 2. Overview of experimental flow for the main study. Specific assessment details are further outlined in Table 3.**

| Session 1 Atlanta VAMC | Session 2 Emory University Hospital | Session 3 Atlanta VAMC |
|---|---|---|
| **All Participants** | **All Participants** | **Stroke Participants** |
| **Cognitive Function:** | **Healthy Older and Younger Adults:** | **Post-Stroke Participants:** |
| Delis Kaplan Executive Function (DKEFS) Trail Making test (A & B) | Pre: MRS (GABA) / Motor Skill Acquisition | Pre: Adapted Motor Skill Acquisition (Joystick) |
| Stroop Color-Word Interference | Exercise/Peripheral Lactate | Adapted Exercise/Peripheral Lactate |
| Oral verbal Fluency (letters FASI | Post: MRS (GABA) Motor Skill Acquisition | Post Adapted Motor Skill Acquisition |
| Digit Span | **Post-Stroke Participants:** | |
| N-back Working Memory | Pre: MRS (GABA) | |
| Digit-Symbol Substitution Test | Exercise/Peripheral Lactate | |
| Hopkins Verbal Learning test | Post: MRS (GABA) | |
| **Quality of Life:** | | |
| SF-36 Health Questionnaire | | |
| Godin Leisure Time Activity Questionnaire | | |
| Pittsburgh Sleep Quality Index | | |
| Beck II Depression Inventory | | |
| **Aerobic Fitness:** | | |
| Modified Balke Submaximal VO2 test | | |
| **Upper Extremity Motor Function: Healthy Older and Younger Adults** | | |
| Grip Strength | | |
| Halstead-Reitan Psychomotor test | | |
| Nine-hole Pegboard test | | |
| Purdue Pegboard test | | |
| Timed Coin-rotation task (uni/bimanual) | | |
| **Upper Extremity Function: Post-stroke Participants** | | |
| Fugl-Meyer Assessment | | |
| Wolf Motor Function Test | | |
| Box and Block Test | | |

return for Session 3, lasting 60 minutes. After arrival, participants engage in a motor skill acquisition task using a joystick. Upon completion of the task, subjects engage in a session of spin exercise for 20 minutes. After exercise completion, participants immediately perform a similar motor skill acquisition task using a joystick.

**Primary outcome.**   The primary outcomes for this study are the change in MRS-assessed GABA levels and motor performance measures related to procedural motor learning (accuracy and reaction time) across all participants. In healthy adults these data are acquired within the MRI (described below) while this is assessed outside the magnet in a follow-up session (Session 3) with persons with stroke.

## Data collection

**Session 1: Consent and behavioral data collection.**   In session 1, the participants can converse with an experienced study coordinator about the study, ask questions, discuss potential

**Table 3. Session 1 behavioral inventories by group.**

| Cohort | Battery | Assessments (domain) |
|---|---|---|
| All participants | Delis-Kaplan Executive Function (DKEFS) | • Trails A & B (processing speed)• Stroop Color-Word interference (processing speed/switching/inhibition) • FAS (letter and semantic fluency) |
| | Working memory | • Hopkins Verbal Learning test (HVLT) (verbal fluency) • Digit span/Reverse Digit Span (memory span) • Symbol-Digit Lookup (processing speed • N-back (reaction time/working memory) |
| | Quality of Life Survey and General Patient Reported Inventories | • Rand Short-Form 36 (SF-36): (general health) • Godin Leisure Time Activity Questionnaire (LTAQ: (physical activity engagement) • Beck Depression Inventory (BDI-II): (affective inventory) • Apathy Scale: (engagement, anhedonia) • Pittsburgh Sleep Quality Index (PSQI): (recent sleep history) • Epworth Sleepiness Survey (ESS): (daily fatigue) |
| Healthy Younger / Older | Upper extremity function | • Halstead Reitan Finger Tapping: (psychomotor speed) • Simple Reaction Time Test: (reaction time) • Nine Hole Pegboard: (dexterity) • Purdue Pegboard: (distal dexterity) • Grip Strength: (hand and pinch grip strength) • Coin Rotation Task: (interdigital dexterity and bimanual response interactions) |
| Stroke | Upper extremity function | • Upper Extremity Fugl-Meyer (Performance based impairment index) • Box and Blocks (unilateral gross motor dexterity) • Wolf Motor function Test (quantitative functional motor ability assessment) • Adapted serial reaction time task (SRTT) using joystick (reaction time and implicit motor skill acquisition) |

concerns, and provide informed consent. Once the participant agrees to take part in the study, they will undergo a battery of motor and cognitive tests summarized in Table 2, previously shown to be sensitive to changes in aging [49–51].

We will obtain an objective assessment of cardiovascular fitness (VO2 max) using a modified Balke protocol (walking) with a Cosmed metabolic cart (see below). This data will be compared with exercise self-report using the physical activity readiness questionnaire (PARQ) for consistency.

Maximal Treadmill Exercise Test: Participants will undergo maximal treadmill exercise until exhaustion using the modified Balke protocol to determine maximal oxygen uptake (VO2 max). The protocol consists of 2-minute stages in which speed remains constant and incline is progressively increased in a stepwise manner. Healthy participants begin walking at a specified rate (1.3 meters per second—or slower if presenting hemiparesis or gait impairment) and 0% incline at the outset of the protocol with grade increasing by 2.5% every 2 minutes. Participants continue the test until failure (maximum tolerable output) or request for cessation. The test is safe for the included age groups and participants with a history of stroke (Billinger et al., 2014). Breath-by-breath levels of mixed expired oxygen, carbon dioxide, and ventilation are recorded at rest and during exercise using a metabolic cart (Cosmed (Rome, Italy)). Breath-by-breath data is smoothed, and VO2 max is the highest VO2 rate observed during maximal exercise. Heart rate will be monitored using the Polar Tracker heart rate monitor. The test is discontinued if the participant reports significant fatigue or exhibits respiratory distress beyond expected exercise exertion as determined by a study exercise physiologist. The

testing is done in a medical center with a nurse available should participants have medical concerns (i.e.- Code Blue). The test is done with adequate ambient ventilation. To minimize droplet transmission, an electrostatic filtration cartridge is fitted to a silicon rubber mask placed over the nose and mouth of the participant.

**Session 2: Measuring brain changes in response to a single bout of exercise using MRS.** Upon arrival at the MRI, study staff will denote timing of procedures during imaging session on a score sheet (see S1 File). Time of acquisition will be recorded for placement in the magnet bore, MRI scan acquisition, measurements of heart rate (before, during and after exercise), lactate recording, and exercise intervals.

Participants will visit Emory University's Center for Systems Imaging Core for MR acquisition during the second session using a Siemens Prisma Fit 3T magnet. Participants will be asked to refrain from consumption of caffeine, nicotine, and psychoactive drugs for 24 hours prior to the imaging session. Radiofrequency (RF) transmission is accomplished via the body coil, and RF reception is accomplished with a 32-channel phased array head coil. Due to concerns about respiratory virus transmission, all participants will be provided with surgical face masks with metal nose clip removed by the study staff. The participant will be required to wear a mask for the imaging session. Participants will view a 1024x768 screen projected via a first surface mirror system through the rear of the bore of the magnet. A Current Design (Philadelphia, PA) four-button inline button box will be secured via medical tape to the participant's torso or right thigh after matching finger position with the appropriate response button. An Avotec (Stuart, FL) Silent Scan 3300 MR-compatible audio system will be used to communicate with participants in the MRI from the console room. Ambient scanner noise will be significantly attenuated by providing all participants earplugs and the Silent Scan clamshell earphones. All participants will verify the capability of hearing scanning staff with a verbal report. Participants will be reminded to remain as still as possible after initial placement in the scanner bore and between scans to attempt movement artifacts. Foam padding will be placed around the participant's head to increase comfort inside the MRI scanner and reduce the potential for participant movement. Before scanning, all participants will verbally confirm the view of the entire visual field after placement in the bore. In addition, participants will perform abbreviated training runs (described below) for motor tasks to verify task understanding and proper functioning of the button box. Heart rate, respiration rate and pulse oximetry will be recorded using Siemens MR-compatible sensors.

During the session, we will obtain structural images using a T1-weighted Magnetization Prepared radio frequency pulse with RApid Gradient Echo (MPRAGE) scan (3D acquisition, TR = 2530 ms, TE = 3.08ms, TI = 1110 ms, FOV read = 256 mm, Flip angle = 7 deg, $1 \times 1 \times 1$ mm voxel size, 262 slices, Bandwidth = 130Hz/Px, Echo spacing = 8.3 ms). Before acquiring MR spectra, a water calibration scan will be performed using a second-order shim. We will plan the second-order shim by visually locating the hand region (hand knob) of the left primary motor cortex as planned on the MPRAGE [52]. To ensure reliability between acquisition, a screen capture of the voxel placement (i.e.–shim box) in the three acquisition planes will be made. Operators will use the screen capture image to guide the re-placement of the shim box on the subsequent scan after the acute exercise session. (Future studies could include the use of fiducial markers with automated table positioning to maximize overlap of the hand knob across sessions.) To limit bone and CSF inclusion and ensure maximal coverage of hand knob of the primary motor cortex we will use following location specifications. The superior extent of the shim box will be aligned with the most dorsal slice of the left motor cortex in the axial plane using care to limit inclusion dura and CSF if cortical atrophy is present. The center of the shim box in the axial plane encompasses the inverted omega of the hand knob. Visual inspection of the shim box in axial, coronal and sagittal planes will then be used to assess

potential inclusion of bone or ventricles. If necessary, the shim box will be placed slightly more lateral on the primary motor cortex and the shim box will be rotated to limit inclusion of the falx and lateral ventricles. While orthogonal acquisition is strongly preferred, operators will note the degree of rotation and attempt to match during the subsequent MRS session after acute exercise. During voxel placement after acute exercise, the previous session's screen capture will be compared to the active shim to maximize overlap. Any deviation in voxel overlap for spatial extent and/or tissue type will be reported as a covariates for repeated measure metabolite comparisons. We will then measure and adjust the shim to account for inhomogeneities in the MR $B_0$ using manual linewidth adjustments on the Siemens console. The Siemens interactive shim tool will be used to adjust the shim to be applied to all MRS exam card parameters. Acceptable linewidth at acquisition will be set to <19 ppm full width at half maximum (FWHM). Manual gradient adjustments will be made in the X, Y and Z plane, as necessary using continuous sampling. After adjustment to the best linewidth estimation, we will apply frequency adjustments in the appropriate Siemens tab until obtaining a delta of 0Hz. We will then acquire resting MR spectra of GABA in the acquisition voxel using a Mescher-Garwood Point Resolved Spectroscopy (MEGA-PRESS) sequence optimized for the Emory scanning environment by an expert physicist (LCK). Each MEGA-PRESS acquisition (TR = 1900ms, TE = 68ms, 30mm x 30mm x 25mm) will last 6 minutes with 128 averages. Additional scan parameters are listed in S2 File. Quality control of the acquired spectra will be reviewed on the Siemens console to review the width of the water peak throughout scanning. Significant changes in the amplitude of the sampled spectra may result in repeating the shim procedure and subsequent acquisitions at the investigator's discretion. MRS spectra will be exported in the Siemens TWIX data format with all dimensions (radio frequency channels and transients) preserved without modification. Water and metabolite data are stored as separate files.

Younger and older healthy participants will complete a functional MRS (fMRS) paradigm involving three conditions: rest, a 12-movement motor learning (adapted serial reaction time task), a button press task, and a recall task. Tasks will be programmed in PsychoPy 2.0 [53] and presented on a 1024x768 screen. The motor sequence learning task is a simplified version (12 movements instead of 16) of the learning task paradigm as implemented by Kolasinski et al. (2019) [54] and performed using the dominant hand (Kolasinski et al., 2019 [54]). A Current Designs (Philadelphia, PA) four-button inline response pad will record button presses in the MRI. Visual cues consist of the outline of a cartoon hand on-screen with colored dots appearing above the target (See Fig 1 for Illustration).

Target fingers will alternate between the index finger and little finger with no immediate finger repeats within the sequence. For example, the target sequence may be 1-4-3-4-2-3-4-2-1-3-1-2, with 1–4 representing the index finger to the little finger, respectively. Digits will be equally represented through the sequence, and the sequence will differ in pre and post-exercise acquisitions. Participants will be instructed to respond as quickly and accurately as possible upon target cue without anticipating presses with the instructions, "You will see the outline of hand on the screen. During the task, the fingertips will light up; your task is to press the button that corresponds to the finger that lights up. You may or may not see a sequence of the fingers that light up. Again, your task is to press the button that corresponds to the finger that lights up." After confirmation and execution of a single sample sequence, participants will begin the task. Errors are denoted as incorrect button presses within the display duration of a target stimulus (1-second). A performance epoch is three successive 12-movement sequences separated by a 4-second rest. After each 12-movement sequence, accuracy will be calculated as a percentage correct and visually displayed to the participant during the 4-second inter-sequence rest (Blackwell & Newell, 1996). Performance epochs are separated by a rest period

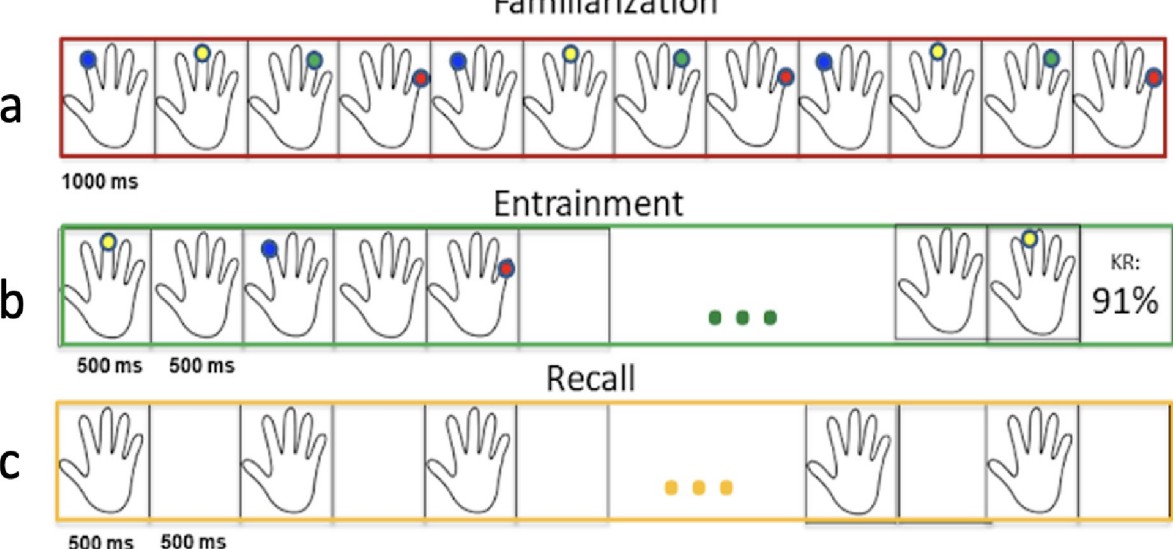

**Fig 1.** Illustration of fMRS stimuli for healthy older and younger adults for the adapted serial response button press task: a) Task familiarization (practice) outside of magnet with target finger press presented for 500ms; b) Entrainment protocol of paced adapted serial response time task with 12 movements; at completion of trial block, knowledge of results (KR) is given to participants for performance feedback; c) Recall task stimuli paced similarly to entrainment (a blank flashing hand provides timing cue for button presses within trial block).

of 20 seconds. A total of 48 sequence iterations are presented within one run. Two motor-sequence learning task runs will be executed before continuing to the motor recall condition. During motor recall, the same pacing of stimuli is presented as in the motor learning condition; however, instead of dots appearing above targets, the flashing of the hand provides the timing cue to participants. Accuracy information is recorded, but no feedback is provided to the participant. Inter-sequence and inter-epoch rests are represented by holding the hand outline stationary on the screen. As head movement is a challenge in stroke patients and the fMRS paradigm may be overly fatiguing, we will only acquire resting data in persons with stroke for feasibility.

Before entering the MRI room, participants will be familiarized on the motor sequence paradigms in a room adjacent to the MRI. Training trials will use similar stimuli and pacing as those used during scanner task execution but will be presented on a laptop computer. Instructions will be given to the participant outside the MRI for both the entrainment and recall conditions immediately before practicing each condition. To be placed in the magnet bore, the participant must complete a 12-movement practice sequence at least three times without error (same criteria for error as above). A maximum familiarization time of 5 minutes will be imposed to limit potential interference effects with the scanner paradigm. Participants will repeat this training in the magnet bore for acclimatization to the MR environment.

### Exercise protocol & post-exercise scanning

After the initial MR acquisition, participants will be escorted to an adjoining room in the MR facility, where they will complete a high-intensity interval-based stationary cycling exercise session while wearing a Polar H10 (Kempele, Finland) heart rate monitor. In association with the Polar Team app and Apple iPad (Cupertino, USA), the H10 allows real-time heart rate and intensity zone tracking during exercise. Participants will also wear a study-provided Apple Watch for validation of heart rate. In addition, participants will wear a Cosmed-brand silicone

rubber face mask with an electrostatic antiviral filter. Seat height will be set roughly equal to the greater lateral trochanter of each participant. A maximum of 20 minutes of exercise is prescribed, with the initial 5 minutes as a warm-up period. After 5 minutes, intervals will increase the workload, generating a heart rate intensity from 50% to the age-predicted maximum. Ratings of perceived exertion (RPE) will be acquired throughout the exercise using the Borg rating scale (6 –no exertion at all to 20 –maximal exertion) [55].

The exercise recommendation falls within the guidelines of the American Heart Association and has been deemed safe for performance in persons with stroke [48]. Therefore, exercise will be terminated under the following conditions: completion of the 20-minute protocol, syncope or report of light-headedness, or by patient request. The exercise will be completed in a hospital environment under the supervision of an experienced exercise physiologist.

After completion of the acute exercise, participants will return to the MRI room to re-enter the magnet. Using the previously acquired MR prescription geometry, we will maximize the voxel overlap of MRS acquisition in the second session. All participants will undergo a resting MRS evaluation of the dominant sensorimotor cortex within 40 minutes of the exercise. Healthy participants will then complete an fMRS protocol using an alternate motor sequence, and stroke participants will undergo a resting MRS evaluation of the non-dominant sensorimotor cortex.

## Peripheral lactate acquisition

Peripheral blood lactate levels will be acquired via finger stick to measure skeletal muscle response to the prescribed exercise load. All operators and study staff will review a standardized protocol developed by the study team for the acquisition to increase reliability of acquisition (see S3 File). Lactate will be sampled at 5-minute intervals during Session 2 (MR acquisition). Immediately before placing the participant in the magnet for the first MR acquisition, the "pre-exercise MR," two lactate samples will be acquired, separated by five minutes. To minimize participant movement, participants will be given clear instructions along with a lactate sampling after placement in the magnet bore. A damp facecloth will be heated outside of the magnet and placed on the sampled finger for ~30 seconds to promote capillary blood flow. After the pre-exercise MR session, we will again sample lactate before the participant leaves the MRI room. Lactate sampling will proceed at the outset of the 20-minute cycling session and every 4 minutes through the exercise. Immediately after exercise, the participant will be transferred to the MR environment. Lactate will again be sampled in the MRI room before "post-exercise MR" acquisition. Between MRS acquisitions, a study team member will enter the MRI room and acquire samples. We will employ repeated measures ANOVA and subsequent paired t-test comparisons of lactate levels before and after exercise across acquisition timepoints. We will plot the lactate recovery curve after exercise to calculate rate of return to resting lactate levels and perform a between subjects t-test for group comparisons.

All sampling is performed via a lancet on the individual's left index finger's distal phalanx (medial side). Before each sample, the skin will be disinfected using a 70% isopropyl-alcohol preparation pad (Medline Industries, Northfield, IL). When the alcohol is completely dried, the finger will be lanced (Roche Diabetes Care, Inc., Indianapolis, IN), and gentle pressure will be applied around the puncture site until ~.05 ml of blood is produced. This blood sample will be wiped clean using a sterilized gauze pad, and pressure will be immediately applied to produce a new sample. A Lactate Pro 2 (Arkray, Japan) blood lactate test meter and disposable test strip will be applied directly to the pooled sample on the finger. All steps will repeat for every blood draw outside the MR environment. All sanitization and wiping procedures will continue in the MR environment; however, the blood sample of interest will be collected via capillary

transfer using a non-ferrous glass pipette (5uL). This sample will then be measured from an aluminum collection pan within 10 seconds of collection using the Lactate Pro 2 device. A manual for the performance of this assessment is shown in S3 File.

**Session 3: Upper extremity function and motor skill acquisition in chronic stroke survivors.** Participants with chronic stroke will complete standardized assessments to evaluate upper extremity movement. These include the Upper Extremity Fugl-Meyer, Box and Blocks Test, Wolf Motor Function Task, and an adapted version of the serial reaction time task (SRTT) performed with a joystick (to accommodate difficulty in the execution of fractionated finger movements). The joystick task (illustrated in Fig 2) involves the movement of a computer joystick (Thrustmaster, Hillsboro, OR) to square targets on a computer screen at 1024x768 resolution. Targets are positioned at the screen's left, right, top, and bottom and measure 96x96 pixels. Like the button press task in healthy adults, the task is divided into three

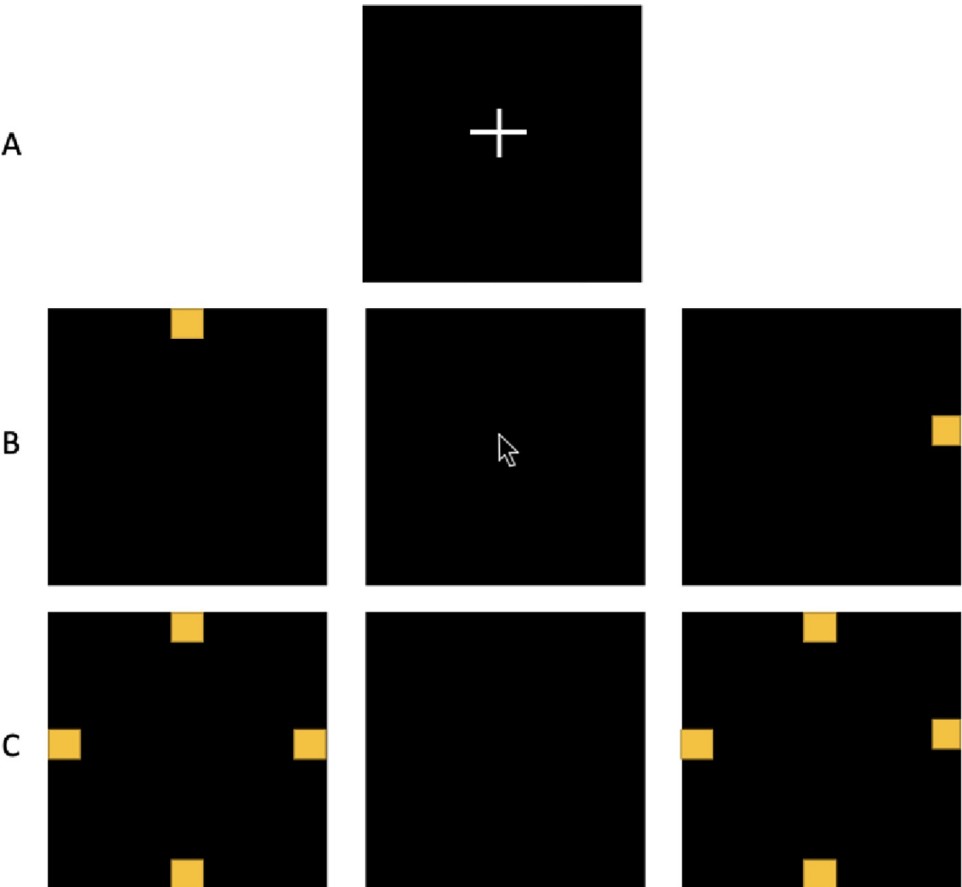

**Fig 2. Illustration of adapted serial reaction time task with joystick in participants with upper extremity impairment after stroke.** A) Participants are asked to remain attentive but relaxed during presentation of a crosshair on the screen (Rest). B) Entrainment task blocks involve presentation of target in location to top, right, left, or bottom of screen for one second. Participants are asked to move the cursor with the joystick to the target during this time. After one second, the target disappears, and the participant is instructed to allow the joystick to return to screen center (home). After 8 movements within a trial block, a crosshair returns to the screen and the participant is instructed to rest. The entrainment sequence repeats 24 times. C) The recall task has the same timing as the Entrainment task, but the participant is not given positional cues (all four positions appear simultaneously). The participant is asked to move the cursor to a target in sequence if they remember identifying a sequence during the task execution. Cursor position is tracked throughout the entirety of the session. Movement accuracy (hitting target or moving in target cardinal direction) and reaction time is measured throughout entrainment and recall sessions.

components: 1) practice training, 2) entrainment, and 3) recall. During entrainment (and practice), the joystick remains at the screen center (home position) until stimulus targets appear in a repeated positional sequence. A crosshair will appear at the screen center at the beginning of each condition, during which the participant is instructed to rest. Next, stimulus targets appear for 1 second when the participant is asked to move the cursor with the joystick within the target area. After one second, the target disappears, and the participant is asked to return the joystick to the home position. The stimulus target will then appear at a different location for one second, and the participant will move the cursor with the joystick to the target. This pattern repeats within a trial block eight times within a 16-second block. After this time, a crosshair will appear at the screen center, cueing the participant to rest until the next block. Movement accuracy (hitting the target or moving in the target cardinal direction) and reaction time is measured throughout entrainment and recall sessions.

## Power analysis and sample size

Our overarching specific aims are to link changes in GABA physiology that are affected by aging and stroke. The only previous studies that have assessed resting GABA changes in sensorimotor areas using MRS at 3T after acute exercise did so in younger adults [37, 47]. Using effect sizes published by Coxon et al. study (d = 2.67) and a study published by Maddock et al., 2016 (d = 1.35), we conservatively estimated an effect size of d = 1.02 for within-participant response to exercise [37, 47].

## Outcomes

All participants' primary behavioral outcome measures are the reaction time and accuracy during the motor sequence learning and recall tasks in healthy patients during MRI and performance on the joystick motor skill learning tasks in participants with upper extremity chronic stroke taken during a testing session outside of the magnet. Primary physiological outcome measures included MRS data, specifically GABA levels, acquired during scanning. Secondary outcomes measures include behavioral data from assessments (survey, cognition metrics, physiological measures). Data will be collected and analyzed offline using Matlab (Mathworks, Danvers, MA) and SPSS (IBM Inc) software.

## Behavioral task analysis

For healthy adults, behavioral data (reaction time, accuracy, change scores) will be collected using PsychoPy 2.0 and loaded into custom-built Matlab scripts to analyze accuracy and reaction time. We will employ a linear mixed effects model (as implemented by lme4 in R) to compare GABA+ values in a 3x2 (between: groups x within: condition) mixed model. Subjects will be held as a random effect. For within-subjects (time), sphericity will be modeled and corrected using a Greenhouse-Geisser adjustment. We will employ pairwise t-tests for main and interaction effects across variables and use a Bonferroni correction to account for family-wise error rate (FWE). In addition, we will employ linear regression to assess relationships in behavioral performance (condition and change scores). An alpha level (corrected for multiple comparisons) of .05 will be set for all analyses.

## Magnetic resonance spectroscopy analysis

MRS spectra will be analyzed offline after secure transfer to the analysis workstation. We will preprocess the TWIX data to correct eddy currents, motion correction, retrospective removal of frequency and phase drifts, spectral alignment, and nuisance peak removal. The GANNET

toolbox (v. 3.3.2) will be used for data processing and analysis (Mullins et al., 2014 [56]). Water will be used as an internal concentration reference. Each dataset will be frequency- and phase-corrected using spectral registration (Near et al., 2015 [57]) and filtered with 3-Hz exponential line broadening. The area under the edited GABA + signal at 3.0 ppm will be estimated. The editing approach here results in a GABA signal containing some inseparable macromolecules (homocarnosine, histidine) coupled to spins at 1.7 ppm. Therefore, all GABA values will be reported as GABA+. GABA+ and unsuppressed water signals will be modeled using a single Gaussian function with linear baseline parameters and a Gaussian-Lorentzian model per implementation in GANNET. Next, we will determine the contribution of tissue type: white matter, gray matter, and cerebrospinal fluid. We will co-register and segment MRS voxels to the MPRAGE image to do this. After co-registration, we will extract GABA+ levels in gray and white matter against measures of water reference (from Harris et al., 2015 [58]). This procedure will be completed for each of the MRS runs. Signal to noise (SNR) will be evaluated from GANNET output for every MRS run and significant drops (> 3 std dev.) in SNR between runs will be noted for further data quality inspection. In addition, we will evaluate Cramer-Rao lower bound (CRLB) values of GABA+/Creatine (Cr) with 20 as a maximum acceptable limit for any run. Within-session GABA+ values from each MRS run (rest, entrain or recall) will be normalized (divided by) to the session's first resting GABA+ scan. In addition, we will calculate GABA+ dynamic range within each session by subtracting the maximum-minimum GABA+ run values for each session. GABA+/H20 will be used for all statistical analyses. A separate analysis of GABA+/Cr will be performed as an ancillary analysis of metabolite ratio reliability/stability.

Functional MRS (fMRS) changes in GABA+ concentration in healthy adults will be evaluated as a change metric against resting baseline in each session. Significant differences and change in GABA+ levels will be determined using a linear mixed model approach (lme4) with age group (between) and condition—pre/post exercise (within) with subjects held as random in R. Subsequent regression analyses will be employed to evaluate how pre-exercise resting GABA+ level predicts change in post-exercise resting GABA+ and post-exercise GABA + dynamic range. For participants with stroke, we will employ a paired-sample t-test to evaluate differences in pre-exercise resting GABA+ to post-exercise GABA+ levels. We will employ a random-effects repeated measures ANOVA to evaluate laterality differences in left and right sensorimotor GABA+ values before and after exercise holding subjects as random with subsequent paired samples t-tests evaluated to measure potential interaction effects.

## Discussion

This cross-sectional study will measure changes in GABA+ due to the engagement in acute aerobic exercise in the dominant motor cortex as assessed with MRS at rest in samples of younger adults, older adults, and adults with chronic stroke affecting the upper extremity. Our main comparisons will be changes in GABA+ between groups at rest after the acute exercise bout. We will also compare changes in GABA+ concentration between older and younger adults at rest and after engagement in a motor learning sequence task. We hypothesize that acute aerobic exercise will increase GABA+ levels in the sensorimotor cortex and this increase will be associated with better motor learning after exercise. We predict that aging will be associated with decreased concentration and total change in GABA+ levels, which will be correlated with poorer motor learning. Persons with cortical stroke will show lower overall GABA + levels as compared to other groups but will show increase after exercise with concomitant improvement in motor learning. We hypothesize that measures of peripheral lactate levels will be positively correlated with GABA+ concentration in the brain.

With respect to aging, a recent study by Heise et al. (2022) demonstrated that performance on a bimanual movement synchrony task is mediated by GABA concentration differentially by age group [42, 59]. Older adults with higher levels of GABA (in the primary motor cortex could transition between phasic motor states more rapidly than individuals with lower GABA levels in the same age cohort). Lower GABA concentration in younger adults predicted a better ability to transition between states. The authors measured electroencephalography (EEG) during the experiment. They found that GABA concentration was a mediator of age-group differences in phase lag performance comparisons of beta frequency, a frequency previously shown in magnetoencephalography to be sensitive to aging-related changes in motor task performance [42, 59]. Heise et al. (2022) reasonably postulated that adaptable GABA concentration might act as a buffer against aging-related declines (and it is noteworthy that overall GABA levels were lower in older adults as compared to younger adults). However, a limitation of the Heise et al. (2022) study is that the group only sampled GABA at a single time point [59]. By exploring GABA changes during rest to task transitions and across multiple time points, we should demonstrate for the first time that GABA concentration changes influence speed of motor learning. In addition, the change in GABA levels post-acute exercise offers a significant opportunity to evaluate the potential mechanistic role of GABA with respect to functional changes associated with motor skill performance [60].

The present study will also evaluate GABA concentration changes before and after exercise in persons with chronic stroke affecting the upper extremity. The only study to evaluate GABA concentration changes in the motor cortex before and after therapy reported that GABA levels were significantly lower in persons with chronic stroke compared to healthy controls at baseline [19]. After two weeks of constraint-induced movement therapy, changes in GABA levels were significantly correlated with performance on the Wolf Motor Function Test. As such, GABA levels may be sensitive to functional performance after a rehabilitation regimen for chronic stroke.

## Supporting information

**S1 File. Magnetic resonance imaging scan/data sheet.**
(PDF)

**S2 File. Magnetic resonance spectroscopy parameters.**
(DOCX)

**S3 File. Standard operating procedure for lactate measure during magnetic resonance imaging.**
(PDF)

**S4 File. STROBE checklist.**
(DOCX)

## Acknowledgments

We would like to thank our colleagues in the Atlanta Center for Visual and Neurocognitive Rehabilitation (CVNR) for helpful revisions to the initial conceptualization of this work. In addition, we extend sincere thanks to Dr. Jay Alberts, Ms. Medina Bello, Ms. Lisa Calas, Ms. Bryana Whitaker, Dr. Arash Harzand, and Ms. Laura Lang.

## Author Contributions

**Conceptualization:** Keith M. McGregor, Thomas Novak, Joe R. Nocera, Steven L. Wolf, Lisa C. Krishnamurthy.

**Data curation:** Keith M. McGregor, Lisa C. Krishnamurthy.

**Formal analysis:** Keith M. McGregor.

**Funding acquisition:** Keith M. McGregor, Lisa C. Krishnamurthy.

**Investigation:** Keith M. McGregor, Joe R. Nocera, Kevin Mammino, Steven L. Wolf, Lisa C. Krishnamurthy.

**Methodology:** Keith M. McGregor, Joe R. Nocera, Kevin Mammino, Steven L. Wolf, Lisa C. Krishnamurthy.

**Project administration:** Keith M. McGregor, Lisa C. Krishnamurthy.

**Resources:** Keith M. McGregor, Lisa C. Krishnamurthy.

**Software:** Keith M. McGregor, Lisa C. Krishnamurthy.

**Supervision:** Keith M. McGregor, Lisa C. Krishnamurthy.

**Validation:** Keith M. McGregor, Lisa C. Krishnamurthy.

**Visualization:** Keith M. McGregor, Lisa C. Krishnamurthy.

**Writing – original draft:** Keith M. McGregor, Thomas Novak, Joe R. Nocera, Kevin Mammino, Steven L. Wolf, Lisa C. Krishnamurthy.

**Writing – review & editing:** Keith M. McGregor, Thomas Novak, Joe R. Nocera, Kevin Mammino, Steven L. Wolf, Lisa C. Krishnamurthy.

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
