## [Decision Letter · Decision Letter 0]

11 Aug 2023

PONE-D-23-15402

Examination of Acute Spin Exercise on GABA Levels in Aging and Stroke: The EASE Study Protocol

PLOS ONE

Dear Dr. McGregor,

Thank you for submitting your manuscript to PLOS ONE. After careful consideration, we feel that it has merit but does not fully meet PLOS ONE’s publication criteria as it currently stands. Therefore, we invite you to submit a revised version of the manuscript that addresses the points raised during the review process.

We look forward to receiving your revised manuscript.

Kind regards,

Nadinne Alexandra Roman, Ph.D.

Academic Editor

PLOS ONE

Journal Requirements:

Reviewers' comments:

Reviewer's Responses to Questions

**Comments to the Author**

1. Does the manuscript provide a valid rationale for the proposed study, with clearly identified and justified research questions?

Reviewer #1: Yes

Reviewer #2: Yes

Reviewer #3: Yes

2. Is the protocol technically sound and planned in a manner that will lead to a meaningful outcome and allow testing the stated hypotheses?

Reviewer #1: Yes

Reviewer #2: Yes

Reviewer #3: Partly

3. Is the methodology feasible and described in sufficient detail to allow the work to be replicable?

Reviewer #1: Yes

Reviewer #2: Yes

Reviewer #3: No

4. Have the authors described where all data underlying the findings will be made available when the study is complete?

Reviewer #1: Yes

Reviewer #2: No

Reviewer #3: Yes

5. Is the manuscript presented in an intelligible fashion and written in standard English?

Reviewer #1: Yes

Reviewer #2: Yes

Reviewer #3: Yes

6. Review Comments to the Author

You may also provide optional suggestions and comments to authors that they might find helpful in planning their study.

Reviewer #1: This manuscript consisted of valuable study protocol for investigation after stroke and elder subjects.

Reviewer #2: I have no comments. The proposed study is feasible and sound.

However, the authors have failed to provide appropriate IRB approval from the Atlanta VAMC and Emory University Hospital, which contradicts PLOS ONE's submission rules for Study Protocols.

Reviewer #3: Thank you for the opportunity to review the protocol ‘Effects of acute exercise on functional GABA levels’. It is an interesting study and valuable to the field, investigating differences in GABA levels across healthy aging and in stroke and how these change with exercise. I have some concerns about the write up of the protocol, as well as a few methodological comments. I appreciate data collection has already commenced and these may not be able to be implemented. My knowledge on exercise assessments is limited, so I apologise for any naivete in questions around the exercise portion of the protocol.

Major concerns:

- Clarity of aims and links to analysis. My understanding is there are two key aims: 1) the change in GABA levels following exercise within each group, and if they differ between groups and 2) how GABA levels relate to motor acquisition. 2 is divided into multiple parts: for healthy controls, the motor learning is being assessed within the scanner and simultaneously measuring changes in GABA levels using functional MRS, while for stroke patients resting GABA levels are being linked to out of scanner motor learning. This needs to be clearer throughout the manuscript.

Introduction: the final paragraph jumps around in regards to questions and hypotheses.

Methods: The first two paragraphs, beginning page 6, line 22, jumps around a lot and does not clearly link your aims to your design

Primary outcome on page 11 lists change in metabolites levels as primary outcome: this should just be GABA levels. This also is contradicted in the first paragraph on page 22, which lists MRS measures as secondary outcomes. Additionally in this paragraph it describes MRI as an outcome, which is not mentioned elsewhere.

- MRS acquisition and analysis

Much more description is needed in regards to the acquisition parameters and clarification in the analysis is required.

Acquisition: how is the voxel placed on the sensorimotor cortex? What anatomical guidelines are used? How are you ensuring replicability/overlap as mentioned on page 18, lines 12-14. More detail is needed in regards to the MRI/MRS procedure (see Lin et al, 2021, NMR Biomed for reporting of MRS parameters) and how it differs between healthy controls and participants with stroke: it appears the healthy control MRI session must be much longer than the healthy controls.

I have a concern regarding motion across the session, especially given the lactate sampling throughout the MRS acquisition. At minimum, I suggest repeating a brief localiser or other anatomical scan to assess for bulk motion changes throughout the session for scans that have not yet been conducted. Examination of the MRS spectra for any suggestion of motion should also be carefully completed.

Analysis: how will preprocessing be conducted? ‘correct eddy currents, motion correction, retrospective removal of frequency and phase drifts, spectral alignment, and nuisance peak removal’? Within Gannet? The ‘we will extract GABA+ levels in gray and white matter against measures of water reference’? Are you referring to the alpha-correction as per Harris et al, 2015, J Magn Reson Imaging? Please clarify. How will normalization to rest GABA+ levels be done? How is GABA+ dynamic range defined or calculated? The analysis regarding the effects of exercise on GABA+ levels is unclear to me, as is the link for motor learning. Please provide more detail. Will the paired-t-test be conducted for both regions for the participants with stroke? I also recommend considering including a secondary analysis using creatine referenced values to investigate any potential water signal changes and processing NAA levels or some other control metabolite to ensure findings are specific to GABA levels. I recommend keeping tracking of time from exercise until MRS scans, especially as it may take longer to set up the older participants again in the scanner and investigating any potential effects of time delay.

- Exercise test: do participants do both the maximal and submaximal test? Are the outcomes from the exercise tests used? I think a little more detail on the description of speed and incline and how this test administered would be appropriate, or a citation of the methods used.

- Power analysis: the power analysis described only seems to detail the within subject effects of exercise. This needs more detail. Also please describe how you arrived at sample size for different groups, or that it is exploratory pilot.

- The analysis for outcomes from sessions 3 for stroke patients is not described.

- The lactate analysis is not described and the sampling details are unclear. Will lactate be sampled between every MRS acquisition? It may be valuable to include it in the session flow schematic.

- Discussion: the discussion needs restructured, I suggest opening with a clear summary of the questions that will be asked, dedicated paragraphs to expected findings and then a clear summary. There are assertions of expected results that are not presently justified by analysis plan: ‘GABA concentration changes characterize flexibility in the GABA system during task engagement’ and ‘opportunity to evaluate the potential mechanistic role of GABA with respect to functional changes associated with motor skill performance’.

Medium concerns:

- The sedentary individuals is not motivated in the introduction and is only introduced in the eligibility table (table 1).

- It may be valuable to split out Figure 1 into more figures or at least make it larger

- I am unsure if it would be valuable to consider the learning timeframe outside the scanner (the pre-training). Would practice effects come in to this?

- Is the alpha level stated on page 22, line 17, the one that will be Bonferroni adjusted for FWE?

- Would like a little bit of information about how much the healthy controls are told about what they are expected to learn? What if participants do not learn the sequence?

Small concerns:

- Throughout the manuscript, the tense switches between past and future, this should be made consistent

- Abstract, page 2:, line 9, should it be ‘limited BY neural damage, rather than to?

- The two aims are not really addressed in the intro of the abstract, you could decrease the background information a little

- Claiming GABA levels as a possible biomarker of motor learning: are you suggesting it as a mechanism of increased learning or predictive?

- Tense switching: page 4, line 9-10

- Unclear statement: page 4, 11-13: Recent trends in the field emphasize modifying rehabilitation approaches to increase sensitivity to individual variability to promote the reclamation of function and improve quality of life

- Citation needed for page 4, line 14, regarding adjuvant therapies

- Page 4, line 23, link cerebral infarction to stroke (or call it stroke)

- GABA target paragraph, beginning page 4, line 20: needs more narration/explanation

- Suggest replacing ‘advancements’ on page 5, line 14, with ‘studies’ or ‘research’ a

- Page 5 line 20, what is ‘practical’ learning?

- Unclear sentence: Identifying how GABA concentration may covary with behavior during motor learning could enable targeting this neurotransmitter system with interventions that can change GABA concentration to facilitate skill acquisition.

- Page 6, line 9-10: GABA system's flexibility to acquire a skill – it is not the GABA system that is acquiring a skill, please reframe

- The motivation for your participant groups, starting page 6, line 22 is disjointed.

- Participants: clarify healthy control recruitment vs. chronic stroke

- Split up the right hand dominance/left hemisphere stroke cells in the stroke eligibility table (table 1)

- Table 1: definition of ‘active substance abuse’?

- Page 10, line 6: ‘similar’ or the same imaging/MRS protocol?

- Define ‘spin exercise’ for the third session for stroke participants – same as other exercise session?

- Unclear statement, page 12, lines 4-6: ‘We will schedule session two on the same day as session one, but it may be scheduled separately as needed based on participant availability and scheduling preferences.’

- Why are some assessments underlined in Table 3?

- Mismatch of ‘1-2-3-4’ in Figure 1 vs. ‘2-3-4-5’ in text (page 15, line 25) is confusing as a reader: suggest using 1-2-3-4 consistency. Similarly consistency of little finger vs. pinky finger.

- ‘Errors are denoted as incorrect button presses within the display of a target (1 second)’: where is target presented? Unclear sentence

- Reference: Borg 6-20 scale, page 18 and line 5

- Two citations when discussion the Heiss study in the discussion

- Use of ‘magnet’ to describe the MRI scanner is quite casual, suggest replacing it with MRI scanner or similar

- The meaning of entrainment is not interchangeable with training, I suggest training should be used

7. PLOS authors have the option to publish the peer review history of their article (what does this mean?). If published, this will include your full peer review and any attached files.

Reviewer #1: No

Reviewer #2: **Yes: **Mark Mikkelsen

Reviewer #3: No

---

## [Author Response · Author response to Decision Letter 0]

12 Nov 2023

Hello,

We thank the reviewers for the opportunity to improve this work. We have made significant changes to the manuscript and have attached to this resubmission. The critiques were thorough and well-conceived. Our attempts to address the comments are made in-line with the current document.

Thank you and best,

Keith McGregor, PhD

PONE-D-23-15402

Examination of Acute Spin Exercise on GABA Levels in Aging and Stroke: The EASE Study Protocol

PLOS ONE

Dear Dr. McGregor,

Thank you for submitting your manuscript to PLOS ONE. After careful consideration, we feel that it has merit but does not fully meet PLOS ONE’s publication criteria as it currently stands. Therefore, we invite you to submit a revised version of the manuscript that addresses the points raised during the review process.

We look forward to receiving your revised manuscript.

Kind regards,

Nadinne Alexandra Roman, Ph.D.

Academic Editor

PLOS ONE

Journal Requirements:

Reviewers' comments:

Reviewer's Responses to Questions

Comments to the Author

1. Does the manuscript provide a valid rationale for the proposed study, with clearly identified and justified research questions?

Reviewer #1: Yes

Reviewer #2: Yes

Reviewer #3: Yes

2. Is the protocol technically sound and planned in a manner that will lead to a meaningful outcome and allow testing the stated hypotheses?

Reviewer #1: Yes

Reviewer #2: Yes

Reviewer #3: Partly

3. Is the methodology feasible and described in sufficient detail to allow the work to be replicable?

Reviewer #1: Yes

Reviewer #2: Yes

Reviewer #3: No

We have attempted to improve the clarity of the methods and materials section to better promote reproduction of the protocol and subsequent data. 

4. Have the authors described where all data underlying the findings will be made available when the study is complete?

Reviewer #1: Yes

Reviewer #2: No

The Department of Veterans Affairs owns the resultant original data from the project. We will make de-identified copies of the data available on the website osf.io in accordance with the publishing recommendation in the data access protocol of the project. We have clarified this in the Dissemination Policy.

Reviewer #3: Yes

5. Is the manuscript presented in an intelligible fashion and written in standard English?

Reviewer #1: Yes

Reviewer #2: Yes

Reviewer #3: Yes

6. Review Comments to the Author

You may also provide optional suggestions and comments to authors that they might find helpful in planning their study.

Reviewer #1: This manuscript consisted of valuable study protocol for investigation after stroke and elder subjects.

Reviewer #2: I have no comments. The proposed study is feasible and sound.

However, the authors have failed to provide appropriate IRB approval from the Atlanta VAMC and Emory University Hospital, which contradicts PLOS ONE's submission rules for Study Protocols.

We thank the reviewer for this statement and have added the necessary language for ethics board approval. A copy of the IRB Approval letter has been added to the submission, as well. PAGE 4

Reviewer #3: Thank you for the opportunity to review the protocol ‘Effects of acute exercise on functional GABA levels’. It is an interesting study and valuable to the field, investigating differences in GABA levels across healthy aging and in stroke and how these change with exercise. I have some concerns about the write up of the protocol, as well as a few methodological comments. I appreciate data collection has already commenced and these may not be able to be implemented. My knowledge on exercise assessments is limited, so I apologise for any naivete in questions around the exercise portion of the protocol.

Major concerns:

- Clarity of aims and links to analysis. My understanding is there are two key aims: 1) the change in GABA levels following exercise within each group, and if they differ between groups and 2) how GABA levels relate to motor acquisition. 2 is divided into multiple parts: for healthy controls, the motor learning is being assessed within the scanner and simultaneously measuring changes in GABA levels using functional MRS, while for stroke patients resting GABA levels are being linked to out of scanner motor learning. This needs to be clearer throughout the manuscript.

We appreciate the comment and have clarified the methods with respect to the motor learning comparators. See Outcomes, Page 16.

Introduction: the final paragraph jumps around in regards to questions and hypotheses.

We have revised the introduction’s hypothesis summary section to clarify the specific hypotheses in the project. We have added the project’s Specific Aims and associated hypotheses to the text on page 4. 

Methods: The first two paragraphs, beginning page 6, line 22, jumps around a lot and does not clearly link your aims to your design

The reviewer’s comment on adding the specific hypotheses to the project’s Aims was very helpful. This helped the flow of the first component of the Methods section. We have added the project specific aims and hypotheses, which better supports the analysis section.

Primary outcome on page 11 lists change in metabolites levels as primary outcome: this should just be GABA levels. This also is contradicted in the first paragraph on page 22, which lists MRS measures as secondary outcomes. Additionally in this paragraph it describes MRI as an outcome, which is not mentioned elsewhere.

We appreciate the comment and have made the suggested changes to the text's outcome measures. 

- MRS acquisition and analysis

Much more description is needed in regards to the acquisition parameters and clarification in the analysis is required.

Acquisition: how is the voxel placed on the sensorimotor cortex? What anatomical guidelines are used? How are you ensuring replicability/overlap as mentioned on page 18, lines 12-14. More detail is needed in regards to the MRI/MRS procedure (see Lin et al, 2021, NMR Biomed for reporting of MRS parameters) and how it differs between healthy controls and participants with stroke: it appears the healthy control MRI session must be much longer than the healthy controls.

We thank the reviewer for advising us of this. We have added more detail regarding the acquisition parameters including the checklist as recommended by Lin et al., (2021) as an appendix. We have also added more detail of the voxel placement within and across acquisition sessions. Pages 11 and 12. We used anatomical guidelines as outlined by Yousry et al., 1997. I believe the reviewer means the healthy control MRI session was longer than the stroke session. If so, this is correct, the sessions with participants with stroke did not involve motor learning due to the individual differences in fractionated finger movements.

I have a concern regarding motion across the session, especially given the lactate sampling throughout the MRS acquisition. At minimum, I suggest repeating a brief localiser or other anatomical scan to assess for bulk motion changes throughout the session for scans that have not yet been conducted. Examination of the MRS spectra for any suggestion of motion should also be carefully completed.

Our apologies. This information was omitted from the previous version of the manuscript. We have added more detail about how we attempted to limit head motion. Indeed, the lactate acquisition was a concern for our group, so we generated a how-to manual for study staff to ensure repeatability and limit displacement of the participant. We acquired another localizer image during piloting for reference and monitored the GABA spectra for changes at each sampling interval. Considerations included providing clear instruction with positional reference and warming the finger with a heated facecloth to limit vascular constriction. We have also added more detail regarding the analysis parameters implemented in Gannett 3.3.2.

Analysis: how will preprocessing be conducted? ‘correct eddy currents, motion correction, retrospective removal of frequency and phase drifts, spectral alignment, and nuisance peak removal’? Within Gannet? The ‘we will extract GABA+ levels in gray and white matter against measures of water reference’? Are you referring to the alpha-correction as per Harris et al, 2015, J Magn Reson Imaging? Please clarify. How will normalization to rest GABA+ levels be done? How is GABA+ dynamic range defined or calculated? The analysis regarding the effects of exercise on GABA+ levels is unclear to me, as is the link for motor learning. Please provide more detail. 

We have added much more detail in the text in regard to the data processing and timing considerations for the acquisition (see also new Appendix: MRS_Parameters Table).

We largely employed the default processing steps within Gannett 3.3.2, which included:

• Eddy correction (from Mikkelson et al., 2020)

• Spectral registration (from Near et al., 2015)

• Water referenced tissue-correction (from Harris et al., 2015)

In addition, we have added appropriate references in the text for definitions of normalization and dynamic range. 

Will the paired-t-test be conducted for both regions for the participants with stroke? I also recommend considering including a secondary analysis using creatine referenced values to investigate any potential water signal changes and processing NAA levels or some other control metabolite to ensure findings are specific to GABA levels. I recommend keeping tracking of time from exercise until MRS scans, especially as it may take longer to set up the older participants again in the scanner and investigating any potential effects of time delay.

For GABA levels at rest in the dominant motor cortex, we will employ a linear mixed model analysis repeated measures analysis of GABA spectra with groups (younger, older, stroke) as between and session (pre/post) while holding subjects as random. A second-level analyses will be completed to compare dominant and non-dominant hemispheres in participants with stroke. The suggestion of a control metabolite (Cr) is well-met and we have added language to describe this in the text.

The comment about time recording is very astute. We implemented a robust time-keeping regimen to record the start time of all measurements (exercise [heart rate/lactate], imaging [runs/spectral evaluation/lactate draws], and session transitions. We have provided a sample acquisition sheet for these measures for reference.

- Exercise test: do participants do both the maximal and submaximal test? Are the outcomes from the exercise tests used? I think a little more detail on the description of speed and incline and how this test administered would be appropriate, or a citation of the methods used.

Our apologies, we have clarified this in the text. The submaximal exercise test was not performed in this project, and was a contingency concern if participants were contraindicated for maximal testing by a physician. This should have been removed. The exercise in the study refers to two activities: 1) VO2 assessment: the cardiorespiratory evaluation is done on a treadmill in session 1; 2) High intensity interval training (HIIT) cycling is done for the acute exercise in the imaging session. 

- Power analysis: the power analysis described only seems to detail the within-subject effects of exercise. This needs more detail. Also please describe how you arrived at sample size for different groups, or that it is exploratory pilot.

This is a fair critique. The study was powered specifically for within-subjects effects. Group comparisons (younger, older and participants with stroke) for this study was exploratory. 

- The analysis for outcomes from sessions 3 for stroke patients is not described.

Our apologies. We have addressed this oversight.

- The lactate analysis is not described and the sampling details are unclear. Will lactate be sampled between every MRS acquisition? It may be valuable to include it in the session flow schematic.

We have clarified the sampling details for the lactate acquisition. Yes, we will sample lactate between every scan. We added an appendix with information about the procedures in detail. In addition, the study flow is characterized in the project score sheet.

- Discussion: the discussion needs restructured, I suggest opening with a clear summary of the questions that will be asked, dedicated paragraphs to expected findings and then a clear summary. There are assertions of expected results that are not presently justified by analysis plan: ‘GABA concentration changes characterize flexibility in the GABA system during task engagement’ and ‘opportunity to evaluate the potential mechanistic role of GABA with respect to functional changes associated with motor skill performance’.

We have modified the text in the Discussion to provide a clearer summary of expected findings with respect to the project’s hypotheses.

Medium concerns:

- The sedentary individuals is not motivated in the introduction and is only introduced in the eligibility table (table 1).

We have clarified this in the text.

- It may be valuable to split out Figure 1 into more figures or at least make it larger

We have enlarged the image and will upload a scaled version to the site.

- I am unsure if it would be valuable to consider the learning timeframe outside the scanner (the pre-training). Would practice effects come in to this?

We have clarified this in the text. We limited the maximum familiarization time to 5 minutes of task execution.

- Is the alpha level stated on page 22, line 17, the one that will be Bonferroni adjusted for FWE?

I believe the reviewer may have meant the text under Behavioral Task Analysis (last sentence). We have clarified that we meant uncorrected alpha.

- Would like a little bit of information about how much the healthy controls are told about what they are expected to learn? What if participants do not learn the sequence?

We have added the text instructions into the Methods for clarity. 

Small concerns:

- Throughout the manuscript, the tense switches between past and future, this should be made consistent

We have corrected tense shifts, as appropriate.

- Abstract, page 2:, line 9, should it be ‘limited BY neural damage, rather than to?

Yes, thank you for that correction. 

- The two aims are not really addressed in the intro of the abstract, you could decrease the background information a little

We have excised the initial sentences to parse the background.

- Claiming GABA levels as a possible biomarker of motor learning: are you suggesting it as a mechanism of increased learning or predictive?

We are focused on increased learning. The predictive component of the training is intended to be derived from a bottom-up procedural process.

- Tense switching: page 4, line 9-10

We have attempted to correct tense switching.

- Unclear statement: page 4, 11-13: Recent trends in the field emphasize modifying rehabilitation approaches to increase sensitivity to individual variability to promote the reclamation of function and improve quality of life

We have attempted clarification of the text.

- Citation needed for page 4, line 14, regarding adjuvant therapies

We have added a reference to Mang et al, 2013.

- Page 4, line 23, link cerebral infarction to stroke (or call it stroke)

Thank you for that correction. 

- GABA target paragraph, beginning page 4, line 20: needs more narration/explanation

We have attempted clarification of the text.

- Suggest replacing ‘advancements’ on page 5, line 14, with ‘studies’ or ‘research’ a

Corrected.

- Page 5 line 20, what is ‘practical’ learning?

Thank you for that correction. 

- Unclear sentence: Identifying how GABA concentration may covary with behavior during motor learning could enable targeting this neurotransmitter system with interventions that can change GABA concentration to facilitate skill acquisition.

We have attempted clarification of the text.

- Page 6, line 9-10: GABA system's flexibility to acquire a skill – it is not the GABA system that is acquiring a skill, please reframe

Corrected.

- The motivation for your participant groups, starting page 6, line 22 is disjointed.

We have attempted clarification.

- Participants: clarify healthy control recruitment vs. chronic stroke

Corrected.

- Split up the right hand dominance/left hemisphere stroke cells in the stroke eligibility table (table 1)

Corrected.

- Table 1: definition of ‘active substance abuse’?

Corrected to “self-reported”

- Page 10, line 6: ‘similar’ or the same imaging/MRS protocol?

Corrected.

- Define ‘spin exercise’ for the third session for stroke participants – same as other exercise session?

Apologies. This was groupthink/writing. 

- Unclear statement, page 12, lines 4-6: ‘We will schedule session two on the same day as session one, but it may be scheduled separately as needed based on participant availability and scheduling preferences.’

We have attempted clarification.

- Why are some assessments underlined in Table 3?

Apologies. The error has been addressed.

- Mismatch of ‘1-2-3-4’ in Figure 1 vs. ‘2-3-4-5’ in text (page 15, line 25) is confusing as a reader: suggest using 1-2-3-4 consistency. Similarly consistency of little finger vs. pinky finger.

Corrected.

- ‘Errors are denoted as incorrect button presses within the display of a target (1 second)’: where is target presented? Unclear sentence

Corrected/clarified.

- Reference: Borg 6-20 scale, page 18 and line 5

Added reference.

- Two citations when discussion the Heiss study in the discussion

Corrected.

- Use of ‘magnet’ to describe the MRI scanner is quite casual, suggest replacing it with MRI scanner or similar

Adjusted, as appropriate.

- The meaning of entrainment is not interchangeable with training, I suggest training should be used

With respect, the use of entrainment is preferred for reporting consistency in the protocol, as this is the name used for the paradigm on the computer and score sheets.

7. PLOS authors have the option to publish the peer review history of their article (what does this mean?). If published, this will include your full peer review and any attached files.

Do you want your identity to be public for this peer review? For information about this choice, including consent withdrawal, please see our Privacy Policy.

Reviewer #1: No

Reviewer #2: Yes: Mark Mikkelsen

Reviewer #3: No

---

## [Decision Letter · Decision Letter 1]

15 Jan 2024

Examination of Acute Spin Exercise on GABA Levels in Aging and Stroke: The EASE Study Protocol

PONE-D-23-15402R1

Dear Dr. McGregor,

We’re pleased to inform you that your manuscript has been judged scientifically suitable for publication and will be formally accepted for publication once it meets all outstanding technical requirements.

Kind regards,

Nadinne Alexandra Roman, Ph.D.

Academic Editor

PLOS ONE

Additional Editor Comments (optional):

Dear Dr  McGregor, before publising, when the manuscript will be checked also for spelling and grammar, please adress the following requests from revisor.

Reviewer #2: - "Gannett" and "GANNET" should be replaced with "Gannet". Please also provide a URL to the Gannet website alongside the Edden et al. 2014 citation: https://markmikkelsen.github.io/Gannet-docs/index.html

- Note that Gannet does not compute CRLB's; only fit error percentages

- "Mikkelson" should be "Mikkelsen"

Reviewer #4: Thank you for the opportunity to review this manuscript, which aims to compare changes in sensorimotor GABA levels before and after acute exercise in younger and older adults, and in individuals with chronic stroke, to understand how these changes correlate with motor learning and skill acquisition. The proposed methodology is interesting, valid and feasible.

There are some minor typographical errors that I list here:

Page 5 under 'Participants', the first paragraph appears to be redundant and I suspect you meant to delete it during editing. the sentence "Participants will be recruited from the local population using advertisements in local newspapers, websites, and other local media." is repeated verbatim in the following paragraph.

Page 9, the words "protocolwith" need a space

Page 11 line 17, I believe the word 'attempt' is an error and you may have meant 'attenuate' or 'prevent' movement artifacts.

Page 12 line 1 'covariates' should be singular 'covariate'

Reviewers' comments:

Reviewer's Responses to Questions

**Comments to the Author**

1. Does the manuscript provide a valid rationale for the proposed study, with clearly identified and justified research questions?

Reviewer #2: Yes

Reviewer #4: Yes

2. Is the protocol technically sound and planned in a manner that will lead to a meaningful outcome and allow testing the stated hypotheses?

Reviewer #2: Yes

Reviewer #4: Yes

3. Is the methodology feasible and described in sufficient detail to allow the work to be replicable?

Reviewer #2: Yes

Reviewer #4: Yes

4. Have the authors described where all data underlying the findings will be made available when the study is complete?

Reviewer #2: Yes

Reviewer #4: Yes

5. Is the manuscript presented in an intelligible fashion and written in standard English?

Reviewer #2: Yes

Reviewer #4: Yes

6. Review Comments to the Author

You may also provide optional suggestions and comments to authors that they might find helpful in planning their study.

Reviewer #2: - "Gannett" and "GANNET" should be replaced with "Gannet". Please also provide a URL to the Gannet website alongside the Edden et al. 2014 citation: https://markmikkelsen.github.io/Gannet-docs/index.html

- Note that Gannet does not compute CRLB's; only fit error percentages

- "Mikkelson" should be "Mikkelsen"

Reviewer #4: Thank you for the opportunity to review this manuscript, which aims to compare changes in sensorimotor GABA levels before and after acute exercise in younger and older adults, and in individuals with chronic stroke, to understand how these changes correlate with motor learning and skill acquisition. The proposed methodology is interesting, valid and feasible.

There are some minor typographical errors that I list here:

Page 5 under 'Participants', the first paragraph appears to be redundant and I suspect you meant to delete it during editing. the sentence "Participants will be recruited from the local population using advertisements in local newspapers, websites, and other local media." is repeated verbatim in the following paragraph.

Page 9, the words "protocolwith" need a space

Page 11 line 17, I believe the word 'attempt' is an error and you may have meant 'attenuate' or 'prevent' movement artifacts.

Page 12 line 1 'covariates' should be singular 'covariate'

7. PLOS authors have the option to publish the peer review history of their article (what does this mean?). If published, this will include your full peer review and any attached files.

Reviewer #2: **Yes: **Mark Mikkelsen

Reviewer #4: No

---

## [Editor Report · Acceptance letter]

15 Feb 2024

PONE-D-23-15402R1 

PLOS ONE

Dear Dr. McGregor, 

I'm pleased to inform you that your manuscript has been deemed suitable for publication in PLOS ONE. Congratulations! Your manuscript is now being handed over to our production team.

Kind regards, 

on behalf of

Dr. Nadinne Alexandra Roman 

Academic Editor

PLOS ONE